# Purification and Characterization of a Novel Fibrinolytic Enzyme from Marine Bacterium *Bacillus* sp. S-3685 Isolated from the South China Sea

**DOI:** 10.3390/md22060267

**Published:** 2024-06-10

**Authors:** Zibin Ma, Jeevithan Elango, Jianhua Hao, Wenhui Wu

**Affiliations:** 1School of Agriculture and Bioengineering, Taizhou Vocational College of Science & Technology, Taizhou 318020, China; zibinma@163.com; 2Department of Marine Biopharmacology, College of Food Science and Technology, Shanghai Ocean University, Shanghai 201306, China; srijeevithan@gmail.com; 3Center of Molecular Medicine and Diagnostics (COMManD), Department of Biochemistry, Saveetha Dental College and Hospitals, Saveetha Institute of Medical and Technical Sciences, Saveetha University, Chennai 600 077, India; 4Department of Biomaterials Engineering, Faculty of Health Sciences, UCAM Universidad Católica San Antonio de Murcia, Guadalupe, 30107 Murcia, Spain; 5Key Laboratory of Sustainable Development of Polar Fishery, Ministry of Agriculture and Rural Affairs, Yellow Sea Fisheries Research Institute, Chinese Academy of Fishery Sciences, Qingdao 266071, China

**Keywords:** fibrinolytic enzyme, *Bacillus* sp., purification, optimum activity, thrombolysis

## Abstract

A novel fibrinolytic enzyme, BSFE1, was isolated from the marine bacterium *Bacillus* sp. S-3685 (GenBank No.: KJ023685) found in the South China Sea. This enzyme, with a molecular weight of approximately 42 kDa and a specific activity of 736.4 U/mg, exhibited its highest activity at 37 °C in a phosphate buffer at pH 8.0. The fibrinolytic enzyme remained stable over a pH range of 7.5 to 10.0 and retained about 76% of its activity after being incubated at 37 °C for 2 h. The K_m_ and V_max_ values of the enzyme at 37 °C were determined to be 2.1 μM and 49.0 μmol min^−1^ mg^−1^, respectively. The fibrinolytic activity of BSFE1 was enhanced by Na^+^, Ba^2+^, K^+^, Co^2+^, Mn^2+^, Al^3+^, and Cu^2+^, while it was inhibited by Fe^3+^, Ca^2+^, Mg^2+^, Zn^2+^, and Fe^2+^. These findings indicate that the fibrinolytic enzyme isolated in this study exhibits a strong affinity for fibrin. Moreover, the enzyme we have purified demonstrates thrombolytic enzymatic activity. These characteristics make BSFE1 a promising candidate for thrombolytic therapy. In conclusion, the results obtained from this study suggest that our work holds potential in the development of agents for thrombolytic treatment.

## 1. Introduction

Thrombosis may cause acute myocardial infarction or venous thrombus embolisms (VTE), which are major human diseases. VTE is a common cause of premature death and morbidity [1,2]. It is estimated that ~40% of deaths globally are due to cardiovascular disease [3]. VTE in blood vessels is related to three factors: the hypercoagulative function of platelets, damage to vascular endothelial cells, and blood turbulence. However, arterial thrombosis is considered to be induced by atherosclerosis [4]. Fibrinolytic therapy may increase bleeding risk using clinical drugs such as tissue-type fibrinolytic enzyme activators and urokinase-type fibrinolytic enzyme activators. Therefore, they are only used for the worst cases of thrombosis. Low molecular weight materials such as the recombinant variants of tissue-type plasminogen activators (rt-PA) are being studied as candidates to reduce the risk of bleeding [5]. For example, activated protein C has been studied for its fibrinolytic-promoting function and anticoagulant effect [6]. Abciximab is a platelet membrane-specific glycoprotein receptor inhibitor [7]. In addition to increasing the risk of bleeding, it also has other side effects such as the reduction in platelets after aspirin administration [8]. Thrombolytic therapy is commonly used for heart muscle infarction and other thrombolytic diseases [9,10]. However, there remains a need for new thrombolysis drugs that are efficient, specific, and safe, and that have few side effects [11].

Fibrinolytic enzymes play a crucial role in the breakdown of fibrin clots. They are divided into plasminogen activators and plasmin-like proteases, each with a specific mechanism of action. Plasminogen activators convert plasminogen to plasmin, while plasmin-like proteases directly degrade fibrin clots [12,13]. Common clinical plasminogen activators include streptokinase, tissue plasminogen activator (t-PA), and urokinase plasminogen activator (u-PA). Despite their widespread use, these enzymes have limitations such as a short half-life, low specificity for fibrin, high cost, and risk of excessive bleeding [14,15]. Therefore, the search for safer and more effective thrombolytic agents from natural sources is essential for the management of thrombotic diseases like cardiovascular disorders.

Marine microorganisms are an important source of bioactive natural products that have activities such as being antioxidant [16,17,18], anti-inflammatory [19,20], anti-cancer [21,22], enzyme inhibitors [23,24], antifungal [25,26], anti-bacterial, anti-algal, and anti-larval [27]. In recent times, microorganisms found in unusual habitats have emerged as crucial sources for identifying strains that produce fibrinolytic enzymes. Among these, actinomycetes have stood out as significant contributors to the production of fibrinolytic enzymes. A new, safe, and non-toxic fibrinolytic enzyme derived from marine *Streptomyces* sp. P3, isolated from marine soil, has been discovered. Analysis of the enzyme’s characteristics has indicated its potential as a novel option for the development of thrombolytic medications [28]. Additionally, the fibrinolytic enzyme produced by *Streptomyces* sp. SD5, originating from high-temperature hot springs, has demonstrated rapid fibrin degradation, showcasing strong thrombolytic properties [29]. Dhamodharan and colleagues have successfully characterized and purified the fibrinolytic enzyme produced by *Streptomyces radiopugnans* VITSD8, sourced from marine brown tube sponges *Agelas conifera* [30]. This enzyme bears resemblance to the one initially isolated from the Gobi radiation zone by our research group [31]. Undoubtedly, abnormal environments, such as the ocean’s high salinity, high pressure, and polar environment, offer promising prospects for future microbiology and biotechnology studies [32].

Despite their numerous benefits, fibrinolytic enzymes have several drawbacks including low fibrin specificity, a short half-life, allergic reactions, resistance to treatment, and the need for higher therapeutic doses which can lead to an increased risk of bleeding [33,34,35]. Nevertheless, research on discovering new, powerful, and safe fibrinolytic enzymes continues. In this study, we present the isolation and characterization of a new fibrinolytic enzyme, BSFE1, derived from the marine bacterium *Bacillus* sp. S-3685 (GenBank No.: KJ023685) found in the South China Sea. The main goal of this research was to purify and analyze the properties of this novel fibrinolytic enzyme from a marine bacterium in order to develop an effective thrombolytic treatment.

## 2. Results

### 2.1. Isolation and Identification of Strain S-3685

A total of 28 strains with fibrinolytic enzyme activities in their culture supernatant were detected in a screening trial using fibrin as substrate. Among these strains, S-3685 showed the highest activity and was selected for further research. A 1422 bp fragment of the 16S rRNA gene of strain S-3685 (GenBank No.: KJ023685) was cloned and sequenced. An alignment of the 16S rRNA gene sequences indicated that the strain belonged to the genus *Bacillus* and the closest relative was *Bacillus safensis* with 95% 16S rRNA gene sequence identity. Also, scanning electron microscopy (Figure 1) revealed rod-shaped bacteria, typical of the genus Bacillus. The strain accordingly was designated *Bacillus* sp. S-3685. The tree ID is based on a maximum parsimony analysis of the 16S rDNA sequences. The 16S rDNA sequences were retrieved from the NCBI database (rRNA/ITS) following blast search using the NCBI Nucleotide Blast tool and aligned using ClustalW, respectively. The phylogenetic tree (Figure 2) was obtained by using MEGA 6.05 software.

### 2.2. Purification of Fibrinolytic Enzyme

The enzyme was purified by ammonium sulfate precipitation, anion exchange chromatography, and finally gel-filtration. In Figure 3, the results of the Q-Sepharose column chromatography show that the unbound proteins had no enzyme activity. An NaCl solution was used for gradient elution, and the results were as shown. Anion exchange chromatography could effectively separate the target protein from pigment and other heteroproteins. The optimum concentration of NaCl for eluting and collecting the target protein was 0.55 M, Figure 4. 

Samples collected through a Q-Sepharose Fast Flow column were used for dextran gel filtration chromatography. Three peaks of UV absorbance were eluted after dextran gel filtration chromatography, among which the activity of the target protein was mainly concentrated in the first peak, which was collected and concentrated for SDS-PAGE in Figure 3 and other property experiments. In Table 1, the purification fold of the target protein was 1.08, with a specific activity of 736.4 ± 1.5 U/mg, and the yield was 25.21%.

### 2.3. Optimum Reaction Temperature and Temperature Stability

The results are shown in Figure 5. The enzyme activity was highest at 37 °C and remained at 92.0% at 30 °C. After mixing the purified enzyme with the substrate, the optimal reaction temperature of the purified enzyme was determined at different temperature points within the range of 10–60 °C. The thermal stability trial showed that the residual activity of the enzyme was relatively stable when stored for a while in the range of 20~45 °C. When the temperature rose to 55 °C, the hydrolysis activity of the enzyme decreased rapidly.

### 2.4. Optimum Reaction pH and pH Stability

The hydrolytic activity and the residual activity of the fibrinolytic enzyme under different pH conditions were measured. As indicated in Figure 6, the optimal pH of the fibrinolytic enzyme hydrolysis is 8.0 and this fibrinolytic enzyme has a good residual activity in the range of pH 7.0~10.0, indicating that the enzyme is stable under alkaline conditions.

### 2.5. Kinetics Parameters of Fibrinolytic Enzyme

The Michaelis–Menten equation is V = V_max_[S]/(K_m_ + [S]). The Michaelis constant K_m_ can approximately reflect the affinity between the enzyme and the substrate catalyzed. The Michaelis–Menten constant K_m_ and V_max_ were calculated from a Lineweaver–Burk plot. The K_m_ and V_max_ values of the purified fibrinolytic enzyme for the substrate fibrin were 2.1 μM and 49.0 μmol min^−1^ mg^−1^ at 37 °C. 

### 2.6. Effect of Different Ions on Fibrinolytic Enzyme Stability

The effect of various ions on fibrinolytic enzyme stability was investigated. Among the various metal ions tested, Na^+^, Ba^2+^, K^+^, Co^2+^, Mn^2+^, Al^3+^, and Cu^2+^ (at concentrations of 1 mM) upregulated the activity of the purified fibrinolytic enzyme. In contrast, Fe^3+^, Ca^2+^, Mg^2+^, Zn^2+^, and Fe^2+^ (at concentrations of 1 mM) reduced fibrinolytic enzyme activity to 96.9%, 92.0%, 98.6%, 95.3%, and 92.6% of the control values, respectively (Table 2).

## 3. Discussion

It is essential to explore abundant microbial resources in unique habitats. Recent research has shown that microbes found in extreme environments such as marine environments, hot springs, and deserts produce a wide range of functional enzymes and natural metabolites, which can be valuable for industrial enzyme development and the discovery of new drugs [36,37]. Despite the challenges of surviving in hyper-arid regions, these areas have been found to harbor a rich diversity of microbes and functional enzymes [38,39]. Zhou et al. previously discovered a fibrinolytic enzyme called Velefibrinase (with a molecular weight of 32.3 k) from marine *Bacillus Velezensis* z01 and evaluated its effectiveness for therapeutic purposes in vivo [40]. Several other research studies have also documented the isolation of fibrinolytic enzymes from various marine origins, including *Bacillus flexus* [41], *Pseudomonas aeruginosa* KU1 [42], and *Aspergillus versicolor* ZLH-1 [43]. 

In this present study, the crude enzymes were purified through three processes: ammonium sulfate fractional precipitation, anion exchange chromatography, and dextran gel filtration. In the literature, different purification steps such as a tangential flow membrane filtration system, dialysis, column chromatography, and HPLC have been reported [40,44] based on the strains used for enzyme isolation and the purification steps have greatly influenced the biological properties of enzymes. The purification process resulted in a 4-fold reduction in the total protein compared to the total protein in the culture broth. According to SDS gel electrophoresis, only one strong protein band was observed following the third purification step, i.e., a ~42 kDa protein affiliated with the fibrinolytic activity. Despite the ~4-fold purification of a 42 kDa protein deduced as the fibrinolytic enzyme, the purification of an active fibrinolytic enzyme was only 1.08-fold, indicating loss of activity through the purification procedure. Further improvements to the procedure or means to stabilize the protein are therefore necessary to enhance the enrichment of the active fibrinolytic enzyme.

In the current investigation, the isolated enzyme displayed its highest level of reaction activity at a temperature of 37 °C. Furthermore, it maintained 92.0% of its activity even at a slightly lower temperature of 30 °C. These findings align with previous studies that have also reported an optimal temperature of 37 °C for the BpKJ-31 enzyme derived from *Bacillus licheniformis* KJ-31 [45], 39 °C for *Marinobacter aquaeolei* MS2-1-derived fibrinolytic enzymes [46], 38 °C for *Streptomyces radiopugnans* VITSD8-derived fibrinolytic enzymes, and 40 °C for Velefibrinase [40]. However, the optimum temperature of our enzymes was lower than that of *Serratia marcescens* subsp. *sakuensi* (43 °C) [47], *Chlorella vulgaris* (45 °C) [48], *Pseudomonas aeruginosa* KU1 (~50 °C) [42], *Shewanella* sp. IND20 (55.5 °C) [49], and *Arthrospira platensis* (72 °C) [50], and higher than that of *Bacillus subtilis* JS2-AprEJS2 (24 °C) [51], *Bacillus subtilis* HQS-3 (26 °C) [52], *Bacillus pumilus* BS15 AprEBS15 (27 °C) [53], and *Bacillus subtilis* ICTF-1 (28 °C) [54].

As indicated in Table 3, the K_m_ and V_max_ values of BSFE1 for the substrate fibrin were 2.1 μM and 49.0 μmol min^−1^ mg^−1^ at 37 °C. Compared to other fibrinolytic enzymes, the results indicate that the fibrinolytic enzyme displayed a certain affinity for fibrin in this study.

The activity of fibrinolytic enzymes from marine organisms is dependent on the presence of divalent metal ions such as Zn^2+^, Mg^2+^, Ca^2+^, Hg^2+^, or Co^2+^ [63,64]. Some studies propose that metal ions like Na^+^, K^+^, and Ca^2+^ can activate these fibrinolytic enzymes [65,66]. While numerous papers have examined the impact of metal ions on crude protease extracts from different sources, only a handful of authors have successfully isolated pure forms of fibrinolytic enzymes [52,54,67,68,69]. In this present study, the stability of the fibrinolytic enzyme was tested by treating it with different metal ions and it was concluded that the enzyme activity was significantly altered by these metal ions. For instance, the enzyme activity was slightly upregulated by Na^+^, Ba^2+^, K^+^ Co^2+^, Mn^2+^, Al^3+^, and Cu^2+^, and, on the other side, downregulated by Fe^2+^, Fe^3+^, Ca^2+^, Mg^2+^, and Zn^2+^. In an earlier study, Velefibrinase activity isolated from marine *Bacillus velezensis* Z01 was increased by Mg^2+^ and Ca^2+^ and decreased by Zn^2+^, Mn^2+^, and Co^+^ [40]. These findings suggest that the metal inhibitory effect is influenced by the specificity of the protease used. To support this concept, the activity of fibrinolytic enzymes (serine metalloprotease) derived from *Serratia marcescens* subsp. *sakuensi* [47] is dependent on divalent metal ions, specifically Mg^2+^, Mn^2+^, and Zn^2+^. In contrast, the activity of fibrinolytic enzymes from *Arthrospira platensis* [50] and *Chlorella vulgaris* [48] is regulated by Fe^2+^.

## 4. Materials and Methods

### 4.1. Reagents and Instruments

The fibrin substrates were purchased from Sigma (San Francisco, CA, USA); Q-Sepharose Fast Flow and Sephacryl^®^ S-100 gel were purchased from Amersham (Boston, MS, USA); Bio-Rad Mini III was purchased from Bio-Rad (Hercules, CA, USA); AKTA-FPLC was purchased from Amersham (Boston, MS, USA); SHIMADZU UV-2550 was purchased from Shimadzu (Kyoto, Japan); ORION Model 818 pH Meter was purchased from Thermos Orion (Shanghai, China). All other analytical grade reagents were obtained from Sinopharm Chemical Reagent Co., Ltd. (Shanghai, China). Ultrapure water was used throughout this study. The enzymatic activity was quantified by measuring the diameter of the lysis cycle on the fibrin plate surface.

### 4.2. Separation and Purification

#### 4.2.1. Isolation and Identification of Strain S-3685

The seawater samples were collected from the South China Sea. They were diluted and spread on a fermentation medium agar plate (10 g glucose, 10 g bean flour, 5 g beef extract, 5 g yeast extract, 0.2 g MgSO_4_, 1 g KH_2_PO_4_, 2 g Na_2_CO_3_, and 20 g agar in 1 L distilled water, pH 7.0). The plates were incubated at 30 °C for three days to form the detectable colonies. Fibrinolytic activity of at least 200 strains was monitored using a fibrin medium [70] (Fibrin-AGAR plate: Fibrinolytic activity of each strain was monitored using a fibrin plate assay according to Jespersen [70] with slight modifications). The fibrin agarose gel (5-mm thick) contained 1.5% agarose, 0.12% (*w*/*v*) fibrinogen, 0.5 U/mL thrombin, and a 0.1 M phosphate buffer (pH 7.4) containing 0.15 M NaCl. The clot was allowed to set for 30 min at room temperature. A well with a 3-mm diameter was prepared using a cock borer. Supernatants (40 μL) were carefully placed into the well. An amount of 10 μL of plasmin (0.72 U/mL) was used as a positive control and 40 μL of Y_m_ broth was used as a negative control. The filled plates were incubated at 37 °C for 24 h and subsequently stained with 0.1% Coomassie Brilliant Blue R-250, 40% methanol, and 10% acetic acid for 1 h and destained in 25% ethanol and 10% acetic acid. The clear zone around the well indicated fibrinolytic activity and was measured using a Vernier caliper. The 16S rRNA gene was amplified according to the method described by Wang et al. 2013 [26]. The obtained 16S rRNA gene sequence was searched and aligned with its closely related sequences retrieved from the NCBI database (rRNA/ITS) following blast search using the NCBI Nucleotide BLAST (National Center of Biotechnology Information, Bethesda, MD, USA). Multiple sequence alignments were obtained using NetApp F840c Cluster w/ (56) 144 GB in Bioedit (Sangon Biotech, Shanghai, China) and the phylogenetic tree was constructed with the MEGA 6.05 software (Arizona State University, Tempe, AZ, USA).

#### 4.2.2. Preparation of Sample

One liter of culture medium of a marine bacterium *Bacillus* sp. S-3685 was centrifuged at 10,000× *g*/min for 5 min. Then the supernatant was taken out and ammonium sulfate at 45% of its saturation was added to salt out for 2 h. The sample was centrifuged at 10,000× *g*/min for 5 min. Then the precipitation was taken out and dissolved in 2 times the volume of ultrapure water for ultra-filtration desalination and it was freeze-dried into crude enzyme powder (all the above experiments were conducted at 4 °C). The crude enzyme powder was stored at −20 °C, and the dry powder was dissolved in phosphate buffers (30 mmol/L, pH 8.0) in the purification process to make a sample with a concentration of 0.1 g/mL.

#### 4.2.3. Purification by Ion Exchange Chromatography

In this study, a slightly modified fibrin plate method of Jespersen et al. [70] was used to determine the activity of plasminase. A Q-Sepharose Fast Flow (210 mL, 30 cm × 3.0 cm) anion column was balanced with a phosphate buffer (30 mmol/L, pH 8.0). The detection wavelength of the UV detector was 280 nm, the flow rate of the mobile phase was 5.0 mL/min, and the injection volume was 40 mL. The enzyme activity of proteins unbound by the Q-Sepharose Fast Flow column after loading was determined. The samples (40 mL, 0.1 g/mL) were loaded, and the unbound protein was washed off with a 30 mmol/L, pH 8.0 phosphate buffer. Then, the phosphate buffer (30 mmol/L, pH 8.0, and containing 0~0.8 mol/L NaCl) was used for linear elution, and the active part was collected. After the combination of high enzyme activity components and ultra-filtration desalination, the sample was used for SDS-PAGE electrophoresis analysis.

#### 4.2.4. Dextrin Gel Filtration

A Sephacryl^®^ S-100 (145 mL, 60 cm × 1.8 cm) gel filtration column was pre-balanced with a phosphate buffer (30 mmol/L, pH 8.0) containing 0.1 mol/L NaCl. The active sample obtained by Q-Sepharose FF was loaded, and elution was carried out with the same buffer solution at a flow rate of 2.0 mL/min. Eluents were collected step by step for analysis.

### 4.3. Optimum Reaction Temperature and Temperature Stability

The temperature of the catalytic reaction system was set in the range of 10~60 °C and a point was set at every 10 °C. The pH of the reaction system was 8.0. After 8 min of reaction, the hydrolysis rate of the enzyme to the substrate was measured. The activity of the enzyme was calculated and the enzyme activity was compared at different temperatures. When the optimum temperature was measured, the enzyme activity at the optimum temperature ±5 °C and ±2 °C was measured again. The enzyme activity at the optimum reaction temperature was set to 100%. Then, the relative value of the enzyme activity was calculated at other temperatures. 

Determination of thermal stability of enzyme: The enzyme solution was incubated at different temperatures with 10 °C intervals up to 70 °C for 30 min. Subsequently, the remaining enzyme activity was assayed at 37 °C and pH 8.0. The hydrolysis rate of an enzyme to the substrate was determined in a reaction system of 37 °C and pH 8.0 after the enzyme solution was kept for 30 min. The residual activity of the enzyme stored at 0 °C was set to 100%. Then, the relative residual activity of the enzyme treated at different temperatures was compared [60,61].

### 4.4. Optimum Reaction pH and pH Stability

Determination of optimum reaction pH: The reaction system adopted a buffer with the different pH values mentioned above, and the pH was set in the range of 3.0~12.0, with the pH interval of different systems being 1.0. The reaction was conducted at 37 °C for 10 min, followed by the determination of the hydrolysis rate of the fibrinolytic enzyme to the substrate under different pH conditions, and the activity of the fibrinolytic enzyme was calculated. The activity of the fibrinolytic enzyme at pH 8.0 was set as 100%, and the relative activity of the fibrinolytic enzyme at other pH values was calculated.

Determination of pH stability: The fibrinolytic enzyme was dissolved in different buffer liquid systems at pH 2.0~12.0, respectively, and stored at 4 °C for 24 h. Then, the remaining activity of the fibrinolytic enzyme stored at different pH values was determined at 37 °C and pH 8.0. The fibrinolytic enzyme activity was set to 100% at the pH condition with the highest residual enzyme activity, and the relative residual activity of the fibrinolytic enzyme was calculated under different pH preservation conditions [60,61].

### 4.5. Effect of Different Ions on Fibrinolytic Enzyme Activity

To examine the effects of different metal ions on fibrinolytic enzyme activity, the enzyme was incubated for 2 h at 4 °C with various metal ions (NaCl, BaCl_2_, KCl, FeCl_3_, CoCl_2_, CaCl_2_, MnCl_2_, MgCl_2_, AlCl_3_, ZnCl_2_, CuCl_2_, and FeCl_2_) at a final concentration of 1.0 mM. The fibrinolytic enzyme activity was measured at the optimal pH 8.0 and 37 °C. The fibrinolytic enzyme activity assayed in the absence of metal ions was defined as the control.

### 4.6. Kinetic Parameters of Fibrinolytic Enzyme

The fibrinolytic enzyme activity was determined by the method described by PARK et al. [25]. The K_m_ and V_max_ values of the purified fibrinolytic enzyme were calculated using fibrin as substrate, assuming that the reactions followed simple Michaelis–Menten kinetics. An aliquot of 0.1 mL fibrinolytic enzyme (final concentration was 17 U/mL) was added to 1.5 mL substrate solution and the mixture was incubated for 10 min in a shaking water bath at 37 °C. One unit (U) of enzyme activity was defined as the amount of enzyme required for the liberation of 1.0 µmol FDP (fibrin degradation products) per minute under the assay conditions.

### 4.7. Data Analysis

Results were expressed as averages of a mean of three independent samples with standard deviation. The data were analyzed by one-way analysis of variance (ANOVA) using Microsoft Excel 2010 (Redmond, WA, USA). The post hoc mean separations were performed by Duncan’s multiple-range test at *p* < 0.05.

## 5. Conclusions

The fibrinolytic enzyme purification process was established by precipitating ammonium sulfate with 45% saturation at 0 °C. The enzyme was then dissolved in a phosphate buffer (30 mmol/L, pH 8.0) and subjected to multi-step purification through Q-Sepharose Fast Flow anion exchange and Sephacryl^®^S-100 gel filtration chromatography. This resulted in a single plasmin band on SDS-PAGE, with a yield of 25.21% and a specific activity of 736.4 ± 1.5 U/mg. Enzyme property analysis revealed that the optimal reaction temperature and pH for the fibrinolytic enzyme are 37 °C and 8.0, respectively. The enzyme exhibited good stability within a pH range of 7.0 to 10.5 and temperatures not exceeding 45 °C. The Michaelis–Menten constant and V_max_ were determined from a Lineweaver–Burk plot, with K_m_ and V_max_ values for the substrate fibrin being 2.1 μM and 49.0 μmol min^−1^ mg^−1^, respectively, at 37 °C. These findings demonstrated the high affinity of the fibrinolytic enzyme for fibrin. The enzyme produced by *Bacillus* sp. S-3685 exhibited unique activity compared to other fibrinolytic enzymes, such as special metal properties of BSFE1. The accurate determination of the biochemical properties in this study highlights the potential of this enzyme for use in functional foods or as a clinical thrombolytic agent. 

## Figures and Tables

**Figure 1 marinedrugs-22-00267-f001:**
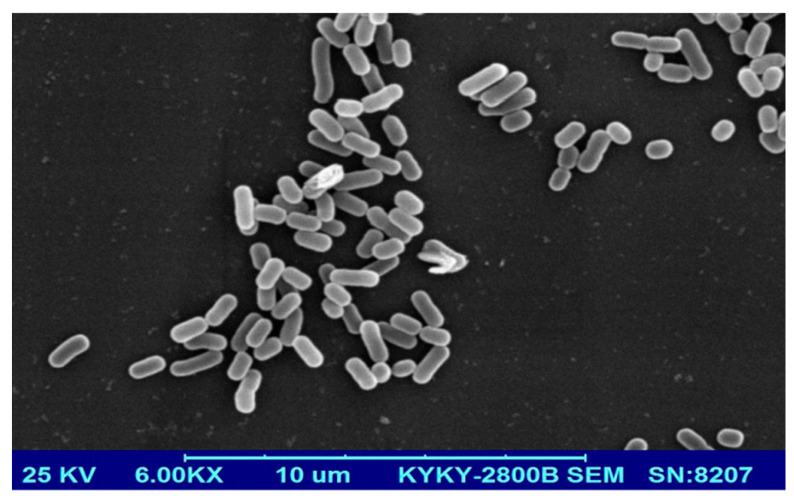
Scanning electron microscope image of strain S-3685.

**Figure 2 marinedrugs-22-00267-f002:**
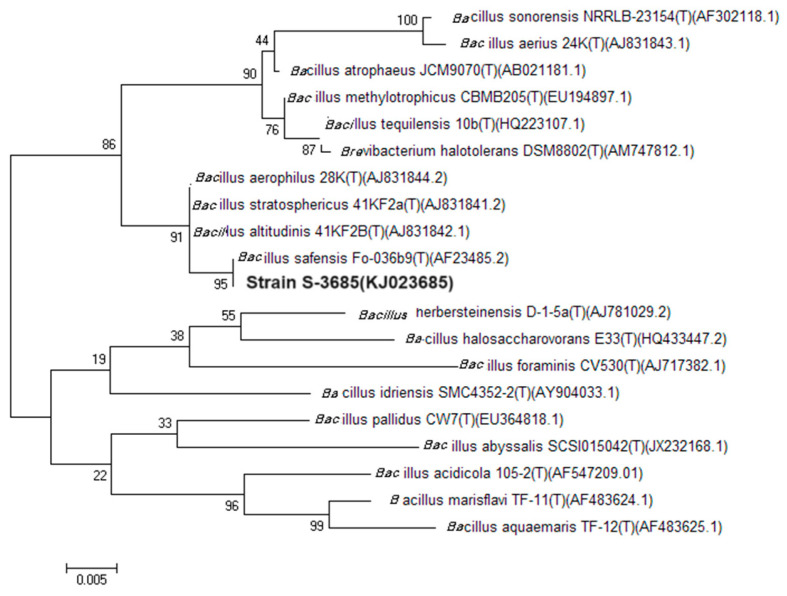
Phylogenetic tree of strain S-3685 and related bacteria.

**Figure 3 marinedrugs-22-00267-f003:**
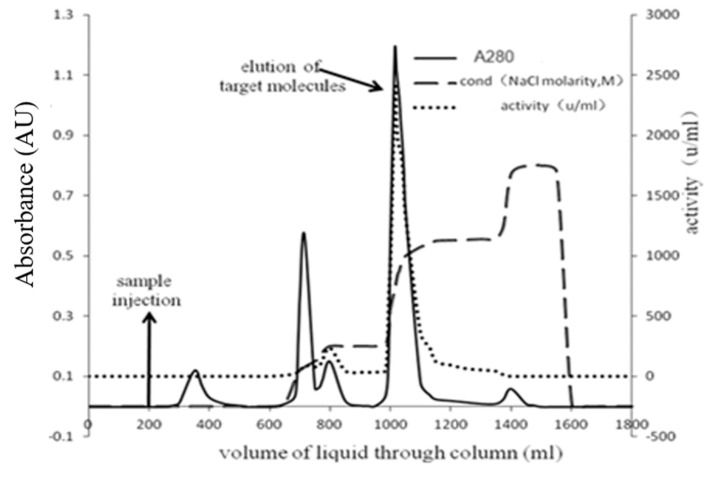
Stepwise gradient elution of enzyme on Q-Sepharose Fast Flow column.

**Figure 4 marinedrugs-22-00267-f004:**
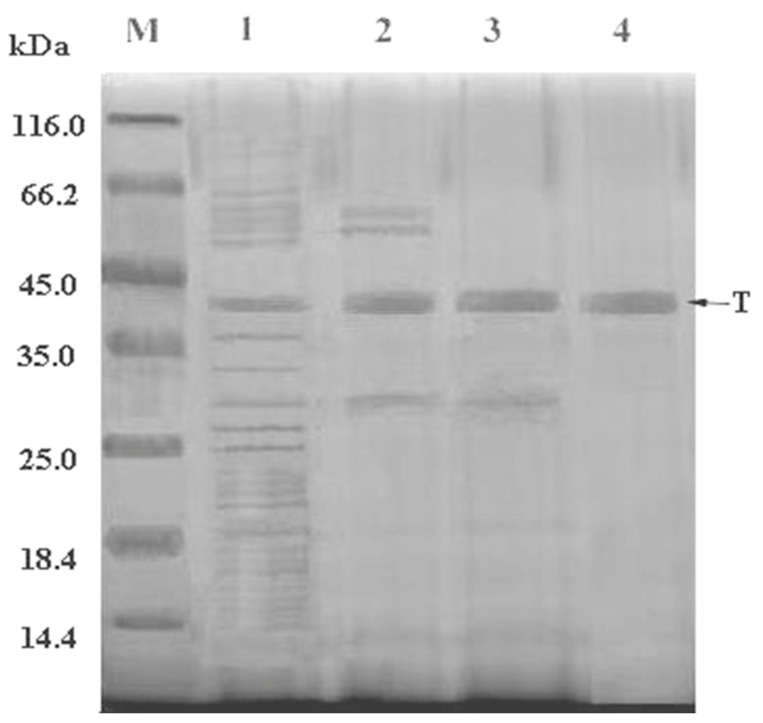
Purification of the enzyme monitored by SDS–PAGE. T, target protein; M, protein marker; 1, culture supernatants; 2, 45% ammonium sulfate precipitation; 3, Q-Sepharose Fast Flow; 4, Sephacryl^®^ S-100 gel filtration chromatograpy.

**Figure 5 marinedrugs-22-00267-f005:**
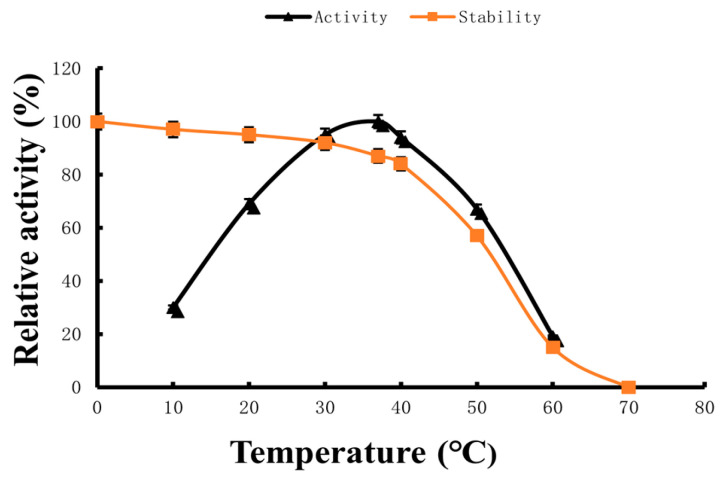
Effects of temperature on fibrinolytic enzyme activity (triangle) and stability (squares). Data are given as means ± SD, *n* = 3.

**Figure 6 marinedrugs-22-00267-f006:**
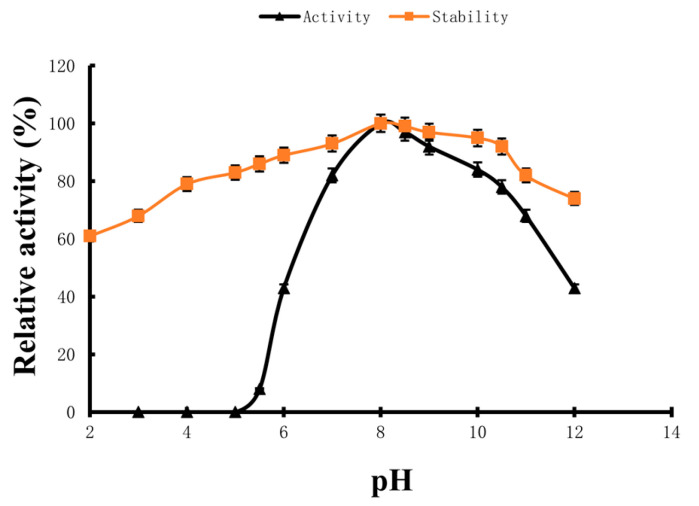
Effects of pH on fibrinolytic enzyme activity (triangles) and stability (squares). Data are given as means ± SD, *n* = 3.

**Table 1 marinedrugs-22-00267-t001:** Purification of enzyme from 1 L of crude supernatant.

Purification Steps	Total Activity ^A^ (×10^3^ U)	Total Protein ^B^ (mg)	Specific Activity (U/mg)	Yield (%)	Purification Fold
Culture broth	1956.23 ± 10.12	2871.67 ± 9.37	681.30 ± 0.53	100	1
Ammonium sulfate precipitation	1406.38 ± 7.75	2002.53 ± 8.13	702.29 ± 0.78	69.73 ± 2.12	1.03
Q-Sepharose FF anion exchange	832.65 ± 5.06	1153.64 ± 6.31	721.20 ± 1.03	40.18 ± 1.45	1.05
Sephacryl^®^S-100 gel filtration	533.37 ± 1.93	723.75 ± 3.57	736.44 ± 1.51	25.21 ± 1.02	1.08

Note: Values were expressed as mean ± SD (*n* = 3). ^A^ Enzymatic activity was measured in 30 mM phosphate buffer (pH 8.0) at 37 °C for 8 h. ^B^ Protein concentration was measured by BCA (method of using BSA (Bull Serum Albumin) as standard.

**Table 2 marinedrugs-22-00267-t002:** Effect of metal ions on the activity of the fibrinolytic enzyme.

Metal Ions	Relative Activity (%)	Metal Ions	Relative Activity (%)
None	100		
Na^+^ (NaCl)	109	Mn^2+^ (MnCl_2_)	103
Ba^2+^ (BaCl_2_)	107	Mg^2+^ (MgCl_2_)	98.6
K^+^ (KCl)	103	Al^3+^ (AlCl_3_)	105
Fe^3+^ (FeCl_3_)	96.9	Zn^2+^ (ZnCl_2_)	95.3
Co^2+^ (CoCl_2_)	102	Cu^2+^ (CuCl_2_)	102
Ca^2+^ (CaCl_2_)	92.0	Fe^2+^ (FeCl_2_)	92.6

**Table 3 marinedrugs-22-00267-t003:** Kinetics parameters of fibrinolytic enzymes.

Enzyme	K_m_	V_max_
BSFE1	2.1 μM	49.0 μmol min^−1^ mg^−1^
Velefibrinase [55]	18.2 μM	59.5 μmol min^−1^ mg^−1^
Matriptase [56]	30.9 μM	
Cocoonase [57]	2.58 μM	40.9 μmol min^−1^ mg^−1^
Li [58]	0.33 μM	
Chen [59]	59.0 μM	
CS624 [60]	21.8 μM	84.0 μmol min^−1^ mg^−1^
XZNUM 00004 [61]	0.96 mg/mL	
Paecilomyces [62]	17.0 μM	

## Data Availability

The data supporting the findings of this study are available upon reasonable request from the corresponding author.

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
