# Peer review of "Purification and Characterization of a Novel Fibrinolytic Enzyme from Marine Bacterium Bacillus sp. S-3685 Isolated from the South China Sea"

_marinedrugs, 2024, doi:10.3390/md22060267_

Round 1

Reviewer 1 Report

Comments and Suggestions for Authors

The manuscript entitled "Purification and Characterization of a Novel Fibrinolytic Enzyme from Marine Bacterium Bacillus Sp. S-3685 Isolated From South China Sea" by Ma, Elango, Sun, Hao & Wu, is a study report on the screening of the bacteria producing fibrinolytic enzyme from sea water, and purification and brief characterization of the enzyme.

The experimental procedure is sound in general, but the results obtained do not seem carefully processed, or discussed enough, as shown in the following points.

L. 20 abstract section; Tense of the sentences is not suitable as follows:

 L. 20 has been > was

 L. 22 exhibits > exhibited

 L. 23 It has been observed that the fibrinolytic enzyme remains > the fibrinolytic enzyme remained

 L. 25 are > were

 L. 26 is > was

L. 21-22

"molecular weight of approximately 42 kDa" should be "molecular weight of approximately 42 k", or, "molecular mass of approximately 42 kDa".

"Molecular weight" is an absolute number, not be accompanied by Da or kDa.

L. 22; "a specific activity of 736.44 U/mg"

> In such an experimental work, significant figures are 3 - 4 digits.

L. 26;

"49.0 mmol min-1mg-1" seems an very high value for specific activity of enzyme in general; "mmol" could be "micro mol". Since no reference value for other fibrynolytic enzyme is presented, the value is doutful.

L. 28; "the fibrinolytic enzyme expressed in this study" 

> the word "express" is usually used to mean production of protein from corresponding gene in enzymological studies. "isolated" may be more moderate expression in this sentence.

L. 86, 241, 242; "Bacillus Sp. S-3685"

> "Bacillus" should be italic.

L. 241;

"culture solution" should be "culture medium"

L. 242; "45% ammonium sulfate"

> Please define "%". Is this % saturation? or weight / vol ?

L. 106-110;

"The detection wavelength of the UV detector was 280 nm, the flow rate of the mobile phase was 5.0 ml/min, and the injection volume was 40 ml. The enzyme activity of proteins unbound by the Q-Sepharose Fast Flow column after loading was determined. "

> This paragraph should be transferred to 4.2.3. materials and methods section.

L. 110;  "The results showed that the unbound proteins had no enzyme activity."

> "The results" should be replaced by "The results of the Q-Sephrose column chromatography"

L. 124;

"The elution peak of dextran gel filtration was 3" should be "Three peaks of UV absorbance were eluted after dextran gel filtration."

L. 128, 130;

The value of specific activity 2.08 should be 1.08 (= 736.44/681.30). Calculation mistake. This means that the enzyme is purified by 8% compared to crude medium; in other word, most protein (around 90%) of the crude medium was the target one. This is incompatible with the result in Fig.4 lane 1. This may be explained by inactivation of the enzyme during the purification procedure.

L. 168-215 discussion section;

Discussion on the characteristics, especially on specific activity, of the fibrinolytic enzymes so far reported from other sources is completely absent.

L. 159-;

Vmax value of 49.0 mmol min-1mg-1 seems very high, compared to other enzyme activity in general. Using a molecular mass of 42kDa and this Vmax value, a turn over number, kcat, of 34300 /sec is calculated. Please compare kcat, or Vmax, value of this enzyme with that of other fibrinolytic enzymes. The same is the case for Km value. The following statement that "These findings demonstrated the high affinity of the fibrinolytic enzyme for fibrin. The enzyme produced by Bacillus Sp. S-3685 324 exhibited superior fibrinolytic activity compared to other enzymes." (L.323-325 in conclusion section) is not supported by any description in the result section. The Km of 2.1 microM is evaluated as "high affinity" (L.160), but more explanation is required, including comparison with other fibriolytic enzymes as well as other general enzymes.

Other minor points;

L. 98-99, "the genus Bacillus safensis";

>  "genus" is not suitable, rather "stain" or "family".

Fig. 2; strain S-3685 should be in bold letters. Some letters in the figure are partially masked!

Fig. 3; OD280 ---> A280 or Abs280 or Absorbance st 280nm; Since activity was measured one by one point, it should be plotted by points not by continuous line in the figure .

Fig. 4; What is the black band at the bottom of the gel? Is this not small protein? Why the 42kD band is assigned to the target protein?

Table 1; "103U"---> 1000U.

Fig. 5; Temperature  ---> Temperature (ËšC)

Fig. 6; Reletivity activity ---> Relative activity

L. 324-325; "The enzyme produced by Bacillus Sp. S-3685 exhibited superior fibrinolytic activity compared to other enzymes." sounds weak, since comparison with other enzymes is not sufficient.

Comments on the Quality of English Language

Experimental results should be written in the past tense.

Author Response

Dear editors,

We sincerely express our thankfulness to the Editor and all the reviewers for their great effort to improve our manuscript quality. The reviewers’ comments give us a big assistance in order to improve our research knowledge and skills. The manuscript has been revised accordingly, and the detailed corrections are listed below point by point:

Reviewer(s)’ Comments to Author:

Reviewer: 1

Comments to the Author

We sincerely thank the reviewer for their great effort and revision to improve our manuscript standard.

Point 1: L. 20 abstract section; Tense of the sentences is not suitable as follows:

  1. 20 has been > was

Response 1: It has been revised as per reviewer’s suggestion.

Point 2: L. 22 exhibits > exhibited

Response 2: It has been revised as per reviewer’s suggestion.

Point 3: L. 23 It has been observed that the fibrinolytic enzyme remains > the fibrinolytic enzyme remained

Response 3: It has been revised as per reviewer’s suggestion.

Point 4: L. 25 are > were

Response 4: It has been revised as per reviewer’s suggestion.

 Point 5: L. 26 is > was

Response 5: It has been revised as per reviewer’s suggestion.

Point 6: L. 21-22

"molecular weight of approximately 42 kDa" should be "molecular weight of approximately 42 k", or, "molecular mass of approximately 42 kDa".

"Molecular weight" is an absolute number, not be accompanied by Da or kDa.

Response 6: It has been revised as per reviewer’s suggestion.

Point 7: L. 22; "a specific activity of 736.44 U/mg"

> In such an experimental work, significant figures are 3 - 4 digits.

 Response 7: It has been revised as per reviewer’s suggestion.

Point 8: L. 26;"49.0 mmol min-1mg-1" seems an very high value for specific activity of enzyme in general; "mmol" could be "micro mol". Since no reference value for other fibrynolytic enzyme is presented, the value is doutful.

Response 8: It has been revised as per reviewer’s suggestion. "mmol" is "micro mol".

Point 9: L. 28; "the fibrinolytic enzyme expressed in this study" 

> the word "express" is usually used to mean production of protein from corresponding gene in enzymological studies. "isolated" may be more moderate expression in this sentence.

Response 9: It has been revised as per reviewer’s suggestion.

Point 10: L. 86, 241, 242; "Bacillus Sp. S-3685"> "Bacillus" should be italic.

 Response 10: It has been revised as per reviewer’s suggestion.

Point 11: L. 241;

"culture solution" should be "culture medium"

 Response 11: It has been revised as per reviewer’s suggestion.

Point 12:L. 242; "45% ammonium sulfate"

> Please define "%". Is this % saturation? or weight / vol ?

Response 12: It has been revised as per reviewer’s suggestion. 45% ammonium sulfate is its saturation.

Point 13: L. 106-110;

"The detection wavelength of the UV detector was 280 nm, the flow rate of the mobile phase was 5.0 ml/min, and the injection volume was 40 ml. The enzyme activity of proteins unbound by the Q-Sepharose Fast Flow column after loading was determined. "

> This paragraph should be transferred to 4.2.3. materials and methods section.

Response 13: It has been revised as per reviewer’s suggestion.

Point 14: L. 110;  "The results showed that the unbound proteins had no enzyme activity."

> "The results" should be replaced by "The results of the Q-Sephrose column chromatography"

 Response 14: It has been revised as per reviewer’s suggestion.

Point 15: L. 124;

"The elution peak of dextran gel filtration was 3" should be "Three peaks of UV absorbance were eluted after dextran gel filtration."

Response 15: It has been revised as per reviewer’s suggestion.

Point 16: L. 128, 130;

The value of specific activity 2.08 should be 1.08 (= 736.44/681.30). Calculation mistake. This means that the enzyme is purified by 8% compared to crude medium; in other word, most protein (around 90%) of the crude medium was the target one. This is incompatible with the result in Fig.4 lane 1. This may be explained by inactivation of the enzyme during the purification procedure.

 Response 16: It has been revised as per reviewer’s suggestion. We thank the reviewer for their great effort to support our manuscript.

Point 17: L. 168-215 discussion section;

Discussion on the characteristics, especially on specific activity, of the fibrinolytic enzymes so far reported from other sources is completely absent.

 Response 17: It has been revised as per reviewer’s suggestion. We have discussed on specific activity of the characteristics.

Point 18: L. 159-;

Vmax value of 49.0 mmol min-1mg-1 seems very high, compared to other enzyme activity in general. Using a molecular mass of 42kDa and this Vmax value, a turn over number, kcat, of 34300 /sec is calculated. Please compare kcat, or Vmax, value of this enzyme with that of other fibrinolytic enzymes. The same is the case for Km value. The following statement that "These findings demonstrated the high affinity of the fibrinolytic enzyme for fibrin. The enzyme produced by Bacillus Sp. S-3685 324 exhibited superior fibrinolytic activity compared to other enzymes." (L.323-325 in conclusion section) is not supported by any description in the result section. The Km of 2.1 microM is evaluated as "high affinity" (L.160), but more explanation is required, including comparison with other fibrinolytic enzymes as well as other general enzymes.

Response 18: It has been revised as per reviewer’s suggestion. More explanation has added.

Point 19: L. 98-99, "the genus Bacillus safensis";

>  "genus" is not suitable, rather "stain" or "family".

Response 19: It has been revised as per reviewer’s suggestion.

Point 20: Fig. 2; strain S-3685 should be in bold letters. Some letters in the figure are partially masked!

Response 20: It has been revised as per reviewer’s suggestion.

Point 21: Fig. 3; OD280 ---> A280 or Abs280 or Absorbance st 280nm; Since activity was measured one by one point, it should be plotted by points not by continuous line in the figure .

Response 21: It has been revised as per reviewer’s suggestion.

Point 22: Fig. 4; What is the black band at the bottom of the gel? Is this not small protein? Why the 42kD band is assigned to the target protein?

Response 22:  It’s a Sample buffer dye front in gel at the bottom, now we have revised. The band 42 kD was chosen based on enzyme activity.

Point 23: Table 1; "103U"---> 1000U.

Response 23: It has been revised as per reviewer’s suggestion.

Point 24: Fig. 5; Temperature  ---> Temperature (ËšC)

Response 24: It has been revised as per reviewer’s suggestion.

Point 25: Fig. 6; Reletivity activity ---> Relative activity

Response 25: Thanks for pointing these errors. It has been revised as per reviewer’s suggestion.

Point 26: L. 324-325; "The enzyme produced by Bacillus Sp. S-3685 exhibited superior fibrinolytic activity compared to other enzymes." sounds weak, since comparison with other enzymes is not sufficient.

Response 26: It has been revised as per reviewer’s suggestion. Here, we want to show unique activity of BSFE1, such as special metal properties.

The revised manuscript has been resubmitted to the journal. We are looking forward to the positive response. 

Yours sincerely,

Zibin Ma, Jeevithan Elango, Mi Sun, Jianhua Hao and Wenhui Wu

Reviewer 2 Report

Comments and Suggestions for Authors

Ma et al. describe and characterize a fibrinolytic enzyme from a marine bacterium (BSFE1). This enzyme is unaffected by some common metal ions and shows high stability at various temperatures and pH levels. According to authors, these properties seem to make BSFE1 a promising candidate for developing drugs to dissolve blood clots (thrombolytic therapy). Although this work contains some interesting results, some parts of the paper should be improved. Generally, the results and discussion section could be more substantial, and I find the results deficiently presented and scientifically poor. This study might publish after further rewriting and revisions:

Major Revisions

- Microorganisms nomenclature needs to be thoroughly revised. Generally, italics are missed, and capital letters are misused. Special attention should be paid to name properly the microorganism source of BSFE1: Bacillus sp. S-3685.

- The GenBank No. KJ023685 refers to the partial sequence of the 16S ribosomal RNA gene of uncultured Bacillus sp. clone 398. Are Bacillus sp. clone 398 and strain S-3685 the same bacterium? This needs to be clarified. In addition, the use of the Genbank reference in abstract and line 86 is misleading, since this reference does not refer to the microorganism, but the 16S ribosomal RNA gene.

- Section “2.1. Isolation and Identification of Strain S-368” is scientifically deficient. In general, the identification of the strain is not clear and poorly presented. Specifically, in this section:

·        There is nothing said about the isolation of the microorganism (where, how, ecology, etc.), which might be relevant in my view. Include some related info.

·        95% of similarity, after the alignment of 16S rDNA partial sequence gene, is not a criterion to determine if this bacterium belongs to one or another specie.

·        Microscopy cannot be used for microbial identification purposes.

·   The methods used in this section are barely mentioned. They should be included in the Materials and Methods section.

·    The phylogenetic tree should contain the names of bacteria properly written. Besides, strain S-3685 is blocking Bacillus herbersteinensis.

- Section “2.2. Purification of fibrinolytic enzyme” needs some clarifications and rewritings:

·        In line 128 is mentioned that the purification fold is 2.08, but the correct value according to the results presented is 1.08. Then, the question here is if the purification method performed, with a yield of 25% and a purification fold of 1.08, is really working and worthy. Authors should clarify this.

·        This section needs an introductory paragraph explaining the purification strategy.

·        Why is the band around 45 KDa in figure 4 associated with the fibrinolytic activity? This should be better explained. In my view, additional experiments would be required to prove this.

- Regarding figures 4 and 5, related to stability of BSFE1 at different temperatures and pH. How is possible to have stability if the residual activity is zero or not measured? This applies for points in graph with temperature 0 °C, and pH 3, 4 and 5. This needs clarification.

- Enzyme class (metalloprotease, serine protease or serine metalloprotease) and enzyme mode of action (reported in terms of direct or indirect fibrinogen lytic activity) are lacking in this work. In my view, these data would strengthen the characterization of the enzyme.

- The kinetic properties of BSFE1 should be discussed in the Discussion section, compared to similar enzymes in the bibliography.

- Culture media and condition for activity expression should be further described in the Materials and Methods section. In this sense, the selective medium used and assayed for fibrinolytic activity is not described (L232).

Discretionary Revisions

- Although the 16S rRNA gene is also designated 16S rDNA, and the terms have been used interchangeably in scientific literature, I would say “16S rRNA gene" is the standard terminology when discussing bacterial identification, taxonomy, and phylogenetic studies, ensuring precise and unambiguous communication.

Minor Revisions

Figures in pages 4 and 5 have the same number, the 4. From this point, the number of figures should change.

L57: “fibrin clots[12,13] Common” to “fibrin clots [12,13]. Common”.

L66: “enzyme inhibitions” to “enzyme inhibitors”.

L77: “Agelas conifer” to “Agelas conifera”.

L98: “16SrDNA” to “16S rDNA”

Author Response

Dear editors,

We sincerely express our thankfulness to the Editor and all the reviewers for their great effort to improve our manuscript quality. The reviewers’ comments give us a big assistance in order to improve our research knowledge and skills. The manuscript has been revised accordingly, and the detailed corrections are listed below point by point:

Reviewer(s)’ Comments to Author:

Reviewer: 1

Comments to the Author

Point 1: Microorganisms nomenclature needs to be thoroughly revised. Generally, italics are missed, and capital letters are misused. Special attention should be paid to name properly the microorganism source of BSFE1: Bacillus sp. S-3685.

Response 1: It has been revised as per reviewer’s suggestion.

Point 2: The GenBank No. KJ023685 refers to the partial sequence of the 16S ribosomal RNA gene of uncultured Bacillus sp. clone 398. Are Bacillus sp. clone 398 and strain S-3685 the same bacterium? This needs to be clarified. In addition, the use of the Genbank reference in abstract and line 86 is misleading, since this reference does not refer to the microorganism, but the 16S ribosomal RNA gene.

Response 2: Thanks for the comment. Here, Strain cloning means to extract 16S rDNA and PCR, it is the same bacterium.

Point 3: - Section “2.1. Isolation and Identification of Strain S-368” is scientifically deficient. In general, the identification of the strain is not clear and poorly presented. Specifically, in this section:

  • There is nothing said about the isolation of the microorganism (where, how, ecology, etc.), which might be relevant in my view. Include some related info.
  • 95% of similarity, after the alignment of 16S rDNA partial sequence gene, is not a criterion to determine if this bacterium belongs to one or another specie.
  • Microscopy cannot be used for microbial identification purposes.
  •  The methods used in this section are barely mentioned. They should be included in the Materials and Methods section.
  •   The phylogenetic tree should contain the names of bacteria properly written. Besides, strain S-3685 is blocking Bacillus herbersteinensis.

Response 3: The reviewer is right, In this study, we only identified which genus the microorganism belongs to, and if we want to identify the specific strain, we also need to do the specific physical and chemical characteristics of the specific microorganism. The isolation details of the microorganism samples was given now (The samples were collected from the South China Sea). The phylogenetic tree is revised now.

Point 4: - Section “2.2. Purification of fibrinolytic enzyme” needs some clarifications and rewritings:

  • In line 128 is mentioned that the purification fold is 2.08, but the correct value according to the results presented is 1.08. Then, the question here is if the purification method performed, with a yield of 25% and a purification fold of 1.08, is really working and worthy. Authors should clarify this.
  • This section needs an introductory paragraph explaining the purification strategy.
  • Why is the band around 45 KDa in figure 4 associated with the fibrinolytic activity? This should be better explained. In my view, additional experiments would be required to prove this.

Response 4: Thanks for your comment. Yes, it is 1.08, we miscalculated. We have revised accordingly. Now microbial fermentation is not optimized, and the index will rise after optimization. We track it based on enzyme activity, so we identified 42 KDa as the target fibrinolytic enzyme.

Point 5: - Regarding figures 4 and 5, related to stability of BSFE1 at different temperatures and pH. How is possible to have stability if the residual activity is zero or not measured? This applies for points in graph with temperature 0 °C, and pH 3, 4 and 5. This needs clarification.

Response 5: Thanks for your comment. In this study, we took the enzyme relative activity at 0°C as 100%. We measured the enzyme relative activity and stability from pH2 to pH 12 separated 1 or 0.5 each point.

Point 6: - Enzyme class (metalloprotease, serine protease or serine metalloprotease) and enzyme mode of action (reported in terms of direct or indirect fibrinogen lytic activity) are lacking in this work. In my view, these data would strengthen the characterization of the enzyme.

Response 6: Thanks for your revisions. In further study, we will complete these research content. And we're doing that right now.

Point 7: - The kinetic properties of BSFE1 should be discussed in the Discussion section, compared to similar enzymes in the bibliography.

Response 7: Thanks for your revisions. It has been revised as per reviewer’s suggestion.

Point 8:- Culture media and condition for activity expression should be further described in the Materials and Methods section. In this sense, the selective medium used and assayed for fibrinolytic activity is not described (L232).

Response 9: It has been revised as per reviewer’s suggestion. Fibrin - AGAR plate was used as the selective medium.

Discretionary Revisions

Point 9: - Although the 16S rRNA gene is also designated 16S rDNA, and the terms have been used interchangeably in scientific literature, I would say “16S rRNA gene" is the standard terminology when discussing bacterial identification, taxonomy, and phylogenetic studies, ensuring precise and unambiguous communication.

Response 9: It has been revised as per reviewer’s suggestion.

Minor Revisions

Point 10: Figures in pages 4 and 5 have the same number, the 4. From this point, the number of figures should change.

Response 10: It has been revised as per reviewer’s suggestion.

Point 11: L57: “fibrin clots [12,13] Common” to “fibrin clots [12,13]. Common”.

Response 11: It has been revised as per reviewer’s suggestion.

Point 12: L66: “enzyme inhibitions” to “enzyme inhibitors”.

Response 12: It has been revised as per reviewer’s suggestion.

Point 13: L77: “Agelas conifer” to “Agelas conifera”.

Response 13: It has been revised as per reviewer’s suggestion.

Point 14: L98: “16SrDNA” to “16S rDNA”

Response 14: It has been revised as per reviewer’s suggestion.

The revised manuscript has been resubmitted to the journal. We are looking forward to the positive response. 

Yours sincerely,

Zibin Ma, Jeevithan Elango, Mi Sun, Jianhua Hao and Wenhui Wu

Round 2

Reviewer 1 Report

Comments and Suggestions for Authors

Figures may be still improved. For example, in Fig. 3,  vertical axis should be explained with "Absorbance (AU)", and so on.  Letters in Figs. 4 & 5 are too small to distinguish.

Activity in Fig. 3 is indicated by dotted curve, not by separated pointed, thus, not acceptable in strict meaning.

Author Response

Dear editors,

We sincerely express our thankfulness to the Editor and all the reviewers for their great effort to improve our manuscript quality. The reviewers’ comments give us a big assistance in order to improve our research knowledge and skills. The manuscript has been revised accordingly, and the detailed corrections are listed below point by point:

Reviewer(s)’ Comments to Author:

Reviewer: 1

Comments to the Author

Point 1: Figures may be still improved. For example, in Fig. 3, vertical axis should be explained with "Absorbance (AU)", and so on.  Letters in Figs. 4 & 5 are too small to distinguish.

Response 1: It has been revised as per reviewer’s suggestion.

Point 2: Activity in Fig. 3 is indicated by dotted curve, not by separated pointed, thus, not acceptable in strict meaning.

Response 2: Thanks for your suggestion. We want to express different items through different forms of curves, so that readers can more easily recognize the meaning of different curves.

The revised manuscript has been resubmitted to the journal. We are looking forward to the positive response. 

Yours sincerely,

Zibin Ma, Jeevithan Elango, Mi Sun, Jianhua Hao and Wenhui Wu

Reviewer 2 Report

Comments and Suggestions for Authors

The paper has improved as it has been partially revised according to the suggestions proposed. However, although authors claim “It has been revised as per reviewer’s suggestion” for most cases, the revisions have not been implemented in:

Point 3: - In section “2.1. Isolation and Identification of Strain S-368”. Specifically, in this section:

  • 95% of similarity, after the alignment of 16S rDNA partial sequence gene, is not a criterion to determine if this bacterium belongs to one or another specie. In the text (L98) it is said thatS-3685 was assigned to the strain Bacillus safensis”. It is confusing.
  •  The methods used in this section are barely mentioned. They should be included in the Materials and Methods section.

Point 4: - Section “2.2. Purification of fibrinolytic enzyme” needs some clarifications and rewritings:

· This section needs an introductory paragraph explaining the purification strategy.

·   Why is the band around 45 KDa in figure 4 associated with the fibrinolytic activity? This should be better explained. In my view, additional experiments would be required to prove this. – Authors replied: “We track it based on enzyme activity, so we identified 42 KDa as the target fibrinolytic enzyme.” Explain this.

 Point 7: - The kinetic properties of BSFE1 should be discussed in the Discussion section, compared to similar enzymes in the bibliography.

 Point 8: Culture media and condition for activity expression should be further described in the Materials and Methods section. In this sense, the selective medium used and assayed for fibrinolytic activity is not described (L232). – Authors replied: “It has been revised as per reviewer’s suggestion. Fibrin - AGAR plate was used as the selective medium”. The medium is not described in the text or given as a reference.

 Point 9: - Although the 16S rRNA gene is also designated 16S rDNA, and the terms have been used interchangeably in scientific literature, I would say “16S rRNA gene" is the standard terminology when discussing bacterial identification, taxonomy, and phylogenetic studies, ensuring precise and unambiguous communication.

 Point 10: Figures in pages 4 and 5 have the same number, the 4. From this point, the number of figures should change.

Please, reply every specific revision with a comment or indicating the modification implemented in the paper.

Author Response

Dear editors,

We sincerely express our thankfulness to the Editor and all the reviewers for their great effort to improve our manuscript quality. The reviewers’ comments give us a big assistance in order to improve our research knowledge and skills. The manuscript has been revised accordingly, and the detailed corrections are listed below point by point:

Reviewer(s)’ Comments to Author:

 Reviewer: 1

Comments to the Author

Point 1: - In section “2.1. Isolation and Identification of Strain S-368”. Specifically, in this section:

  • 95% of similarity, after the alignment of 16S rDNA partial sequence gene, is not a criterion to determine if this bacterium belongs to one or another specie. In the text (L98) it is said that “S-3685 was assigned to the strain Bacillus safensis”. It is confusing.

Response 1: Thanks for your suggestion. In this study, we only roughly identified which genus the microorganism belongs to, and if we want to identify the specific strain, we also need to do the specific physical and chemical characteristics of the specific microorganism.

Point 2: - Section “2.2. Purification of fibrinolytic enzyme” needs some clarifications and rewritings:

  • This section needs an introductory paragraph explaining the purification strategy.
  •  Why is the band around 45 KDa in figure 4 associated with the fibrinolytic activity? This should be better explained. In my view, additional experiments would be required to prove this. – Authors replied: “We track it based on enzyme activity, so we identified 42 KDa as the target fibrinolytic enzyme.” Explain this.

Response 2: Thank for your suggestion. We track it based on enzyme activity. We performed active electrophoresis, measured the plasminase activity after cutting the glue, and identified the target plasminase approximately 42k.

Point 3: - The kinetic properties of BSFE1 should be discussed in the Discussion section, compared to similar enzymes in the bibliography.

Response 3: Thanks for your suggestion. It has been revised as per reviewer’s suggestion.

Point 4: Culture media and condition for activity expression should be further described in the Materials and Methods section. In this sense, the selective medium used and assayed for fibrinolytic activity is not described (L232). – Authors replied: “It has been revised as per reviewer’s suggestion. Fibrin - AGAR plate was used as the selective medium”. The medium is not described in the text or given as a reference.

Response 4: It has been revised as per reviewer’s suggestion. The reference 62 is given in the text.

 Point 5: - Although the 16S rRNA gene is also designated 16S rDNA, and the terms have been used interchangeably in scientific literature, I would say “16S rRNA gene" is the standard terminology when discussing bacterial identification, taxonomy, and phylogenetic studies, ensuring precise and unambiguous communication.

Response 5: Thank for your suggestion. The bacterial identification was sequenced for 16S rDNA genes because RNA is unstable. The experimental reference method also used 16S rDNA and others did the same.

Point 6: Figures in pages 4 and 5 have the same number, the 4. From this point, the number of figures should change.

Response 6: It has been revised as per reviewer’s suggestion.

The revised manuscript has been resubmitted to the journal. We are looking forward to the positive response. 

Yours sincerely,

Zibin Ma, Jeevithan Elango, Mi Sun, Jianhua Hao and Wenhui Wu